# Superconducting In Situ/Post In Situ MgB_2_ Joints

**DOI:** 10.3390/ma16196588

**Published:** 2023-10-07

**Authors:** Bartlomiej Andrzej Glowacki

**Affiliations:** 1Department of Materials Science and Metallurgy, University of Cambridge, 27 Charles Babbage Road, Cambridge CB3 0FS, UK; bag10@cam.ac.uk or bartlomiej.glowacki@ien.com.pl; 2Institute of Power Engineering, DZE-2, ul. Mory 8, 01-330 Warsaw, Poland

**Keywords:** MgB_2_ material, wires, bulks, critical current density

## Abstract

The superconducting joints of superconducting in situ MgB_2_ wires have been of great interest since the first MgB_2_ wires were manufactured. The necessity of joining fully reacted wires in applications such as NMR brings complexity to the methodology of connecting already reacted wires sintered under optimised conditions via a mixture of Mg + 2B and subsequential second heat treatment to establish fully superconducting MgB_2_ joints. Some of the data in the literature resolved such a procedure by applying high cold pressure and sintering at a low temperature. A topical review publication did not address in depth the question of whether cold sintering is a potential solution, suggesting that hot pressing is the way forward. In this paper, we discuss the potential joint interfacial requirements, suggesting a thermo-mechanical procedure to successfully form a superconductive connection of two in situ reacted wires in the presence of Mg + 2B flux. The critical current at 25 K of the researched junction achieved 50% I_c_ for an individual in situ wire.

## 1. Introduction

The superconducting joints of superconducting in situ MgB_2_ wires have been of great interest [1,2,3] since the first MgB_2_ wires were manufactured [4,5,6]. Persistent current joints between technological superconductors were the subject of a topical review [7] in which the authors discussed in depth the joint manufacture techniques for five major technological superconductors: NbTi, Nb_3_Sn, MgB_2_, Bi_2_Sr_2_Ca_2_Cu_3_O_x_, and RE-Ba_2_Cu_3_O_7−x_. It was suggested by the authors of [3], that the melting point of MgB_2_ is high, and at high temperatures the compound tends to decompose into non-superconducting compounds, so the heating and recrystallising of reacted MgB_2_ wires is not an effective method for forming high-quality superconducting joints. Also, it was indicated that applying high pressure at room temperatures is similarly ineffective, as MgB_2_ is a brittle ceramic, much harder than the metal sheath materials. Both of the above conclusions are debatable and questionable. In the earlier technique of making joints between ex situ MgB_2_ wires via in situ Mg + 2B jointing, presented by [8], ex situ wires inserted into a Mg + 2B chamber did not show superconductivity in a 10 K self-field and showed only very weak superconductivity at 4.2 K when sintered at 640–670 °C; they were non-superconducting if sintered at temperatures below or above this range of temperatures. These disappointing results were due to procedural experimental mistakes, not a principally unachievable rule. 

Instead of cold pressing the Mg + B precursor powder, followed by heat-treatment at ambient pressure, the author of [9] used a hot-pressing technique to produce a dense, filler material. On the other hand, hot pressing may not be most convenient method of making joints in an industrial environment; therefore, using our experience with cold deformation and the densification of joints [1], as well as optimisation expertise in the co-sintering of hybrid MgB_2_ in situ wire in direct contact with an ex situ barrier [10,11,12], in the present study, we attempt, based on the Mg–B binary phase diagram and the formation kinetics of MgB_2_, to define interfacial conditions for the thermo-mechanical formation of fully reacted MgB_2_ wires and secondary higher-temperature sintering in the presence of Mg + 2B.

## 2. Materials and Methods

Single-core in situ wires with an OD of 0.7 mm were manufactured according to a standard in situ procedure, as previously described in detail [13]. It was established that a short sintering time at ~700 °C was sufficient to achieve good performance for the in situ wires used. The architecture of the joint is schematically presented in Figure 1.

After filling the joining chamber with Mg+2B powder mixture (of the same composition as that used for the manufacture of the wires), the joined part of the two wires was fixed in place relative to the capsule (effectively preventing lateral movement of the constituting wires) for protection and to enhance the mechanical strength. For further strengthening and improvement of the density of the joined part, as well as to create localised subchannels for the diffusion of Mg into the already-formed MgB_2_ wire cores and intragranular microcracks (to be explained in the bulk of the text), the joint was uniaxially compressed to a pressure of up to 0.5 GPa perpendicularly to the contacting core-containing zone (Figure 1b). Finally, the joint was inserted into a furnace with an Ar protective atmosphere at ambient pressure for heat treatment at a sintering temperature of 900 °C; the sintering duration was 30 min. Afterwards, the samples were cooled down to room temperature in the furnace with an adjusted dwelling time at 650 °C. The sintering temperature and the duration were chosen according to the Mg–B binary phase diagram, and the heat treatment conditions were chosen according to the in situ PIT prepared wires used here.

Generally, I_c_(T, B) measurements of the joints and wires under investigation are conducted versus the cooling temperature and external magnetic flux density in a dedicated system developed to minimise the use of He under dynamic cooling conditions [14]. The I_c_ defining electric field (E) criterion used was E = 1 μV/cm, both for the length unit of the optimised wire and for the resulting joining distance between the wire ends. For this material development paper, only representative, comparative I_c_ data for the wire and resulting joint are provided at 25 K in a self-field. 

Knowledge of the Mg–B binary phase diagram is very important for the synthesis of superconducting MgB_2_ composites, wires, and joints, especially if complex problems of diffusive formation and re-bonding at the interface between the already-reacted in situ wire and the in situ filler during the second co-sintering need to take place. It is important to realise that in such a case, if the in situ reacted wire is subjected to compression and deformation and the re-establishment of its intergrain MgB_2_ connectivity, it is a problem of re-sintering “ex situ” material in contact with the in situ capsule material; this not only adds nomenclatural difficulties to describe the individual stages of the reactive diffusion processes but, most importantly, also poses complex material science problems to be correctly defined, analysed, and resolved. 

Some of the Mg–B phase diagrams presented in the literature can provide uncertain ranges of decomposition temperatures for magnesium boride phases such as ex situ MgB_2_.

The reported decomposition reaction temperature range of boride phases such as 

2MgB_2_(s) ⇒ MgB_4_(s) + Mg(g) is 850 °C ≤ T_decomp_ (MgB_2_) ≤ 1550 °C [15].

The CALPHAD method and ab initio calculations were conducted by [15] to compute thermodynamic model parameters to determine the decomposition temperatures of magnesium borides and update the Mg–B binary phase diagram (see Figure 2). The planning of the sintering procedure of the “ex situ” (already in situ reacted) wires in the in situ joint capsule was based on the resulting Mg–B binary phase diagram in the present manuscript. It is important to note that based on the findings, the boundary between Mg(s) + MgB_2_ and Mg(g) + MgB_2_, as well as the MgB_2_ + MgB_4_ zone, on the elaborated phase diagram remains unchanged irrespective of the external pressure within the range 0.1–100 MPa.

## 3. Results

Before we come to presentation of microstructural and current transport results, aspects of possible parallel important events taking place must be discussed to underline the influences on the chosen process and results. The thermo-mechanical processes taking place during joint formation at the interface between the in situ sintered lower-density elongated MgB_2_ structure of the wires and the Mg + 2B ⇒ MgB_2_ filler have two fundamental aspects, detailed as follows.

### 3.1. Induced Interfacial Change and Phase Transformation

There is a potential change in phase transformation chemistry at the open ends of the MgB_2_ conductor prepared by angular grinding (to maximise the exposure of the MgB_2_ to a potential joining reaction, which could induce local loss of Mg, thus making it prone to decomposition via 2MgB_2_(s) ⇒ MgB_4_(s) + Mg(g)); see Figure 2. 

As pointed out by [16], the thermodynamic decomposition of MgB_2_ takes place under a low Mg partial pressure [17] via 2MgB_2_(s) ⇒ MgB_4_(s) + Mg(g). Such a decomposition of MgB_2_ was experimentally observed as the loss of gaseous Mg at temperatures as low as 610 °C [18]. Importantly, a sintering experiment conducted on MgB_4_ additives to in situ (Mg + 2B) materials [19] revealed a positive effect of such additions at a small percentage (up to 10 wt%). A MgB_2_ sample made by an in situ process at 775 °C for 3 h with 2 wt% of MgB_4_ powder showed a 40% higher critical current density at 25 K in a self-field than did the pure MgB_2_ sample. However, further increase of the MgB_4_ powder content up to 10 wt% caused a systematic reduction in the J_c_(B,T), but not below the actual value for the pure MgB_2_ sample. The excessive loss of Mg at 900 °C in the unprotected interface causes the rapid formation of MgB_4_, which causes degradation of the superconducting properties [20].

As shown by [16], ex situ samples heated at moderately high temperatures of 900 °C for a longer period showed an increased packing factor, a larger intergrain contact area, and significantly decreased resistivity, all of which indicate the solid-state self-sintering of MgB_2_.

In conclusion, some interfacial MgB_4_ formation can take place, freeing Mg vapour and facilitating the healing of microcracks; however, such a process can be localised and can be reversed by excess Mg originating from the Mg + 2B filler, as will be discussed in Section 3.2. 

### 3.2. Recompaction and Recombination in the Wire–Filler Bond

Considering that the contact area between MgB_2_ grains in the ex situ bulk is limited by the porosity, intergrain coupling is still considered insufficient compared with that in the in situ bulk. Thus far, the connectivity of ex situ MgB_2_ is a trade-off between the higher packing factor and the weaker intergrain coupling. The authors of [16] predicted a high connectivity of 30–40% for moderately sintered ex situ MgB_2_ with a packing factor (PF) of 75%. It was calculated that an approximately 25% increase in PF compared with that in in situ MgB_2_ results in up to three times higher connectivity in ex situ MgB_2_ if a sufficient arrangement of surface contact between grains is achieved, for example, by compressing the matrix to 0.5 GPa, re-deforming the grain structure of the MgB_2_ wires [1]. Such procedure would, under a controlled atmosphere of Mg pressure sourced from Mg + 2B, promote the solid-state self-sintering of MgB_2_ and significant improvements in intergrain coupling by heat treatment under ambient pressure; additionally, the reversible process MgB_4_(s) + Mg(g) ⇒ 2MgB_2_(s) can take place, see Figure 2.

It was reported in the literature [21] that ex situ and in situ samples sintered at 900 °C for 24 h show similar J_c_(0 T, 20 K) values, indicating that exposure of the in situ joint in the presence of ex situ wires is a correct approach to the manufacture of superconducting joints. However, the J_c_ value in the presence of an external magnetic field is superior for the ex situ wires; this is less important for the joints, which are normally shielded from the magnetic field of the application electromagnet, such as MRI magnets. Clear evidence for strongly connected ex situ MgB_2_ polycrystalline bulks fabricated by solid-state self-sintering (at 900 °C) [21] formed a basis for planning the sintering of the researched joints at 900 °C for 30 min and cooling them to 650 °C to allow residual magnesium vapour to penetrate the residual interfacial nonuniformities (Figure 2). 

The partial pressure of the Mg vapour in the closed environment at the interface between ex situ MgB_2_ and in situ Mg + 2B (where the complex process of reactive diffusion formation (in situ) of MgB_2_ accompanied by possible MgB_2_ (ex situ) localised decomposition to MgB_4_, releasing Mg and partially re-forming back to MgB_2_, takes place) is essential for the integrity of the superconducting joint to be formed. The high-temperature vaporisation thermochemistry of the intermediate phases in the B-rich portion of the Mg–B phase diagram from MgB_2_ to MgB_20_ was extensively studied by [22], who provided assessment data for the biphasic region where MgB_4_(s) = MgB_2_(s) + Mg(g).

### 3.3. Results of Thermo-Mechanical Processes Joint Formation

Considering the above discussed necessity of powder compaction of the already formed MgB_2_ material in the wires, a cold uniaxial deformation process of the capsulated joint set-up was adopted (Figure 1b). This rearranged the MgB_2_ grains close to induced cracks, enable better accessibility of Mg vapour to promote diffusion bonding of the wires with Mg + 2B joint filler. The resulting morphology and microstructure of the jointing MgB_2_ materials are presented in Figure 3a. The critical current measurement of the representative final joint in a dynamically cooled flow cryostat at 25 K was 50 A, while for the wire, the value was 100 A. 

## 4. Discussion and Conclusions

The presented positive results reflect an analysis of the possible interfacial and localised complex processes of formation of in situ joints with already manufactured in situ wires. Further analysis of the deformation processes and actual sintering reformation and formation processes based on reactive diffusion processes in the presence of Mg liquid and gas forms requires further research efforts to achieve reproducible joints with greater J_c_ properties. 

It is important to realise that in the proposed case, if the in situ reacted wire is subjected to compression and deformation and the reestablishment of MgB_2_ intergrain connectivity, it becomes a problem of sintering “ex situ” material not strictly in situ (material that, however, was in situ before the reaction), which introduces not only nomenclatural difficulties to describe but also, most importantly, complex material science problems to be correctly analysed and resolved. As reliable and reproducible joints are needed for applications, in-depth research will continue.

## Figures and Tables

**Figure 1 materials-16-06588-f001:**
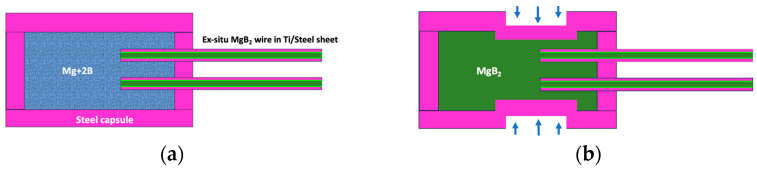
Schematic of the adopted joint architecture: (**a**) assembled in situ reacted MgB_2_ wires inside a capsule filled with pre-compressed Mg+2B mixture of the same composition used for the manufacture of the wires [13]; (**b**) junction from Figure 1a after 0.5 GPa uniaxial compression in the joining area and sintering at 900 °C for 30 min, followed by cooling, as described in the text.

**Figure 2 materials-16-06588-f002:**
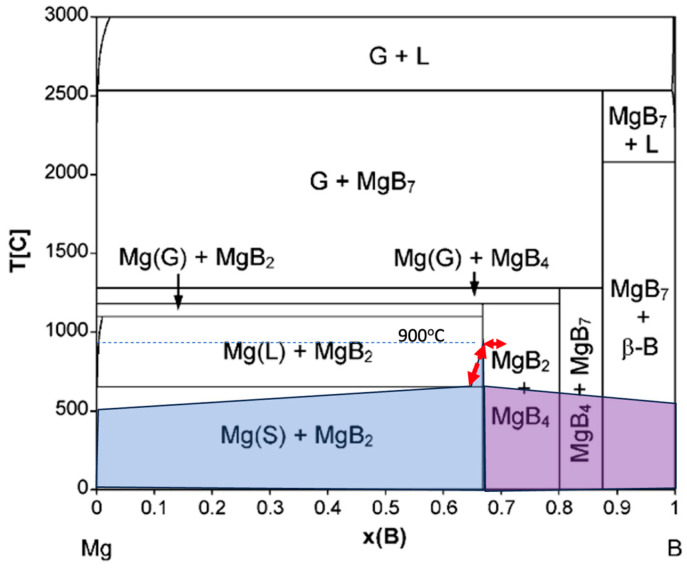
Mg–B binary phase diagram from [15]. The shaded area represents the availability of Mg and B for the filler reaction process after junction compression to 0.5 GPa. The horizontal “reversible” arrow represents discussion of MgB_2_ and MgB_4_ interformation presented in Section 3.1. The vertical “reversible” arrow represents an excess of Mg at the initial stage of sintering of the joints and solid-state diffusion of the joints during cooling presented in Section 3.2.

**Figure 3 materials-16-06588-f003:**
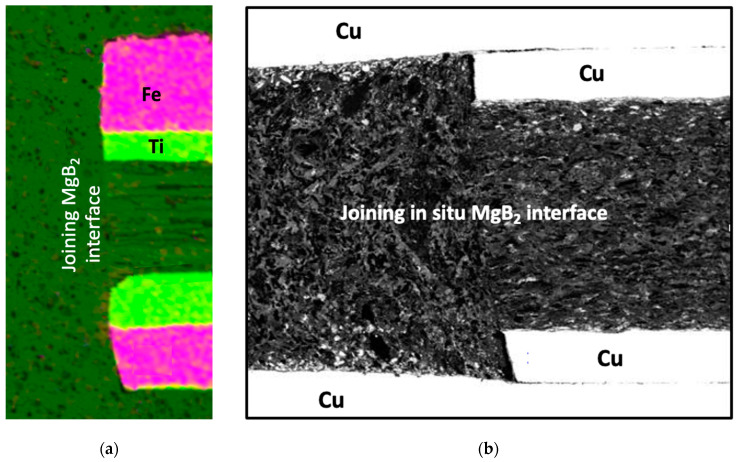
SEM image of the cross-sectional interface between the wire and filler in the joining capsule. (**a**) EDAX current work; the OD of the wire was 0.7 mm. The different morphology of the wire core and the filling material can be observed at the actual joining interface. The fibre-like structure of the original wire induced by cold deformation of the powder is pronounced, whereas the filler matrix looks very uniform. A deep green colour represents MgB_2_. (**b**) For comparison, the cross-sectional microstructure of the initial fully in situ joint formation is presented [1], where electromagnetic compaction created a distinctive buckling effect on the interfacial region between the wire and the filler, which is not observed in Figure 3a. The above comparison of morphological differences of the fibre-like ex situ wire in a rigid in situ capsule and the in situ wire in contact with in situ filler powder reveals the complexity of the deformation processes taking place in various joints. An important fact is that the I_c_ value for the joint presented in Figure 1b was only 15% that for the wire (at 25 K self-field).

## Data Availability

On request.

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
