# Peer review of "Superconducting In Situ/Post In Situ MgB2 Joints"

_materials, 2023, doi:10.3390/ma16196588_

Round 1

Reviewer 1 Report

Manuscript ID: materials-2543319
Title: Superconducting in-situ / post in-situ MgB2 joint

Authors: B. Glowacki

This paper present mostly a discussion about the potential MgB2 joint between two in-situ wires in the presence of Mg+2B flux. Author is writing that the critical current of researched junction achieved 50% of Ic of an individual in-situ wire at 25 K, but there is no experimental evidence confirming it. Generally, there is really luck of adequate experimental results in this manuscript.

Materials and Methods

·       The single core in-situ wires OD 0.7 mm were manufactured ….more details of wire manufacture and its basic properties are needed.

·       The architecture of the joint is not presented clearly and sufficiently ….. the joined part of the two wires was fixed for protection and enhancing the mechanical strength….? What does it mean?

·       It is not clear how the steel capsule is filled by powder….and pressed.

·       Figure 1 shows ex-situ MgB2/Ta/SS joined wires, but this manuscript says about the jointing in-situ ones …?

·       The Ic(T, B) measurements of the joints and wires under investigations were conducted …No Ic(T, B) measurements for joints and wires are shown by this manuscript!

Results

·       Why some theorems are presented as results? It can be possibly used for discussion, but not for results!

·       The resulting morphology and microstructure of the jointing MgB2 materials is presented in Figure 3(a).  Why Figure 3(a) shows MgB2/Ti/SS wire and Figure 1(a) MgB2/Ta/SS joined wires?

·       Figure 3(b) shows the microstructure of the fully in-situ joint of MgB2/Cu wires, which is not comparable with the presented one!

·       The critical current of the final joint at 25K was 50 A where for the wire the value was 100A. No results of measurements are shown! Were the mentioned Ic values measured in self-field? It is really unbelievable to reach 50 % of current carrying capacity for the joint with so small contact area shown by Figure 3(a).

Discussion and conclusions

·       What author means by presented positive results?  This manuscript presents only one experimental result shown by Figure 3(a), which is absolutely not sufficient for scientific paper! 

·        It is true that reliable and reproducible and well characterized joints are needed for presenting superconducting in-situ/post in-situ MgB2 joint, as title says.

Author Response

The Referee is questioning all aspect of the presented methodology and representative results. For the Referee It is unbelievable to reach 50 % of current carrying capacity for the joint with so small contact. Having conducting research for last 50 years on development and characterization of superconducting materials I provided inside knowledge as the reactive diffusion processes of the formation of the ex-situ joints that may take place in the proposed circumstances. The morphology of the in-situ joint was provided for comparison with interfacial morphology of ex-situ joints.

Someone who is working in development of superconducting materials would know that 1 microvolt criterion per cm length of the superconducting conductor (or joint) is the common criterion of Ic.

I do not believe that extensive I(B,T) measurements results are needed at this stage of the joint formation optimization where the main focus is on materials science of the Mg-B phase development and transformation during proposed joint formation. Ic data at 25K is the most representative for LH2 applications.

In my view referee most likely has not follow the explanation of the reactive diffusion formation processes associated with dynamics of MgB2 formation and unique MgB4 phase formation.

In my opinion this short manuscript is an important guidance for other researchers to further improve quality joints as it was rightly pointed by referee It is true that reliable and reproducible and well characterized joints are needed for presenting superconducting in-situ/post in-situ MgB2 joint, as title says.”

I am sorry for not being too responsive to specific demands of the referee, but in my opinion this short manuscript is an important guidance for other younger researchers to further improve quality ex-situ joints with emphasise on materials science.

Of course, I am grateful for pointing out some aspects which may need better explanation or correction for the benefit of potential readers.

Reviewer 2 Report

English is so bad that the content of the manuscript is hard to understand. This is not acceptable since the author is with the University of Cambridge and could certainly find a colleague to help him to improve the manuscript. The objectives of the work are not well explained. It is hard to understand why the author tries to achieve the junction by establishing a superconducting connection between the ends of the wires only, which means that the effective surface of the junction is very small. More interesting results were obtained by other teams as soon as 2010 (see Li et al. doi:10.1088/1742-6596/234/2/022020). I think however that this work could be interesting for researchers in the field if the presentation was strongly improved, the objectives of the work clearly detailed and the more interesting results emphasised. I also demand that the author responds to the remarks and questions in the attached file consequence, I recomment a major revision of the manuscript

English is so bad that the content of the manuscript is hard to understand. This is not acceptable since the author is with the University of Cambridge and could certainly find a colleague to help him to improve the manuscript.

Author Response

Referee comment “English is so bad that the content of the manuscript is hard to understand. This is not acceptable since the author is with the University of Cambridge and could certainly find a colleague to help him to improve the manuscript.” is rather interesting, maybe after 50 years working and publishing development and characterisation of superconducting materials, now I should spend time to learn English…

Referee experienced problem with understanding intention of the manuscript which covers materials science issues of the complex reactive diffusion phase formation and re-formation during proposed ex-situ junction formation.

Referee: “It is hard to understand why the author tries to achieve the junction by establishing a superconducting connection between the ends of the wires only” Junction usually has 2 ends.

I am sorry for not being too responsive to specific demands of the referee, but in my opinion this short manuscript is an important guidance for other younger researchers to further improve quality ex-situ joints with emphasise on materials science.

Of course, I am grateful for pointing out some aspects which may need better explanation or correction for the benefit of potential readers.

Reviewer 3 Report

In the current manuscript, the author presented a detailed review of in-situ growth of  superconducting joint. Over all, I think the novelty and the existing questions upon this topic is well introduced. However, I have some concerns and questions as well. I would recommend its publication only if the author will provide more information I requested and address my concerns.
Major:
1. Correct my if I were wrong but this manuscript is not a review report, meaning that new insightful data need to be shown. I wonder what new data is related to superconductivity in this manuscript. SEM image is only demonstrating information for structure, not directly or indirectly linked to the superconducting behavior which is the main point of this manuscript. More data and spectroscopy of the MgB2 joint need to be added.
2. The second figure is based on ref. 15. I wonder what new information does the author bring in this figure. And the original data for these information need to be added.

Minor:
1. Line 25: BISCCO, REBCO need to be explained.
2. Line 29: room temperature would be more reasonable.
3. Line 33: showed rather than shown
4. Grammar need to be checked through the manuscript.

Author Response

Referee: “I wonder what new information does the author bring in this figure 2”

Referee is questioning the explanation of the process of formation and re-formation of the ex-situ wires in the in-situ joint based on understanding of the reactive diffusion processes which are not fully reflected on Mg-B phase diagram.

As probably referee knows that reactive diffusion processes are only guided by standard phase diagram that are constructed by melting and quenching given composition of binary elements and has very little to do with reactive diffusion kinetic processes conducted at predefined temperatures.

SEM images are provided as a guidance to underline the difference between ex-situ and in-situ deformation induced resulting microstructure.

I am sorry for not being too responsive to specific demands of the referee, but in my opinion this short manuscript is an important guidance for other younger researchers to further improve quality ex-situ joints with emphasise on materials science.

Of course, I am grateful for pointing out some aspects which may need better explanation or correction for the benefit of potential readers.

Reviewer 4 Report

The author reports in this paper, with title "Superconducting in-situ / post in-situ MgB2 joint" (materials-2543319), on the thermo-mechanical procedure to make superconductive MgB2 connection of two in situ reacted wires by means of a Magnesium and Boron flux. In the work, it is claimed that the interfacial conditions are defined according to binary phase Mg-B diagram data and formation kinetics of MgB2. Overall, the methodology and research work are OK and they reach the standard of the journal Materials. So that, my recommendation is to publish the work, but after the following significant revisions:

  1. In the abstract section should not be quoted so many references. Besides, the “Ic” parameter must be defined.

  2. More data should be added to Figure 1 to better illustrate the information. For example, which is the final volume in Figure 1b or what is the difference in volume between Figures 1a and 1b?.

  3. The “Discussion and Conclusions” section is too short, it should be extended, including much more details regarding the conclusions achieved during the investigation of the work.

Moderate editing of English language required.

Author Response

I am grateful for the Referee comments and further expansion and clarification of the three points provided will be implemented to the manuscript in due course.

Round 2

Reviewer 1 Report

Revised version

Figure 1(a) shows ex-situ MgB2/Ti/SS wire used for joint with the reference [13], which is describing the MgB2/Ti/Metal wire made by in-situ process. It means that in-situ MgB2/Ti/SS wire [13] is used for joint.

Author is writing: “The critical current measurements of the representative final joint in dynamically cooled flow cryostat at 25K was 50 A where for the wire the value was 100A.”

But, where is any evidence of it?  It is really very surprising when the author (having conducting research for last 50 years on development and characterization of superconducting materials) is presenting 50 % current carrying capacity for superconducting MgB2 joint without any of clear experimental evidence!

Author is writing: “only representative, comparative data of Ic of the wire and resulting joint are provided at 25 K at self field”. 

Why these “comparative data” are not simply shown in this manuscript?

Author is writing: “I do not believe that extensive I(B,T) measurements results are needed at this stage of the joint formation optimization where the main focus is on materials science of the Mg-B phase development and transformation during proposed joint formation.”

Extensive I(B,T) measurements are not needed, but some of basic or minimal one (e.g. comparison of I-V curves measured for the joint and used wire at the same conditions) are needed.  

His comment to standardly used Ic definition added into the revised version is absolutely not adequate answer on the question of reviewer.

I have to repeat that the luck of adequate experimental results and no evidence confirming superconducting connection between joined MgB2 wires do not allow me to advise this manuscript for publication.

Author Response

Dear referee 1

Thank you for the comments

  • in-situ MgB2/Ti/SS wire was used for joint.
  • Evidence of measurements that 50 % is that I have measured the critical current myself and the repeatability of the measurements of such junction were high and the representative data was provided in Ampers measured with widely accepted standard Electric field criterion.
  • provided absolute data of Ic values are definitely enough for the comparison in this materials science manuscript. If referee does not understand that I-V  characteristic at this stage of junction under development is not a subject of n-value or flux creep definition values for persistent mode operation,  I cannot help him at this stage of junction development. More systematic data of the n-value and the flux creep of the MgB2 junction will follow if improved junction performance in magnetic field will be achieved in future.

Reviewer 2 Report

The author has strongly improved his manuscript which is now easily understandable and interesting for the workers in the field. I recommened publication after, if possible, a moderate editing of the english language.

Although strongly improved the english in the manuscript is not perfect. 

Author Response

Thank you very much for your constructive comments which will definitely contribute to better clarity of the presented procedure of complex MgB2 junctions formation in this important Materials Journal.

Reviewer 4 Report

In my opinion, the revised manuscript has been sufficiently improved to
warrant publication in the journal Materials.

Minor editing of English language required

Author Response

(The authors gave the same response as above.)
